# A Simple Neural Attentive Meta-Learner

**Nikhil Mishra** [*†]      **Mostafa Rohaninejad**[*]      **Xi Chen**[†]      **Pieter Abbeel**[†]

UC Berkeley, Department of Electrical Engineering and Computer Science
Embodied Intelligence
{nmishra, rohaninejadm, c.xi, pabbeel}@berkeley.edu

## Abstract

Deep neural networks excel in regimes with large amounts of data, but tend to struggle when data is scarce or when they need to adapt quickly to changes in the task. In response, recent work in *meta-learning* proposes training a *meta-learner* on a distribution of similar tasks, in the hopes of generalization to novel but related tasks by learning a high-level strategy that captures the essence of the problem it is asked to solve. However, many recent meta-learning approaches are extensively hand-designed, either using architectures specialized to a particular application, or hard-coding algorithmic components that constrain how the meta-learner solves the task. We propose a class of simple and generic meta-learner architectures that use a novel combination of temporal convolutions and soft attention; the former to aggregate information from past experience and the latter to pinpoint specific pieces of information. In the most extensive set of meta-learning experiments to date, we evaluate the resulting Simple Neural AttentIve Learner (or SNAIL) on several heavily-benchmarked tasks. On all tasks, in both supervised and reinforcement learning, SNAIL attains state-of-the-art performance by significant margins.

## 1 Introduction

The ability to learn quickly is a key characteristic that distinguishes human intelligence from its artificial counterpart. Humans effectively utilize prior knowledge and experiences to learn new skills quickly. However, artificial learners trained with traditional supervised-learning or reinforcement-learning methods generally perform poorly when only a small amount of data is available or when they need to adapt to a changing task.

Meta-learning seeks to resolve this deficiency by broadening the learner's scope to a distribution of related tasks. Rather than training the learner on a single task (with the goal of generalizing to unseen samples from a similar data distribution) a meta-learner is trained on a distribution of similar tasks, with the goal of learning a strategy that generalizes to related but unseen tasks from a similar task distribution. Traditionally, a successful learner discovers a rule that generalizes across data points, while a successful meta-learner learns an algorithm that generalizes across tasks.

Many recently-proposed meta-learning methods demonstrate improved performance at the expense of being hand-designed at either the architectural or algorithmic level. Some have been engineered with a particular application in mind, while others have aspects of a particular high-level strategy already built into them. However, the optimal strategy for an arbitrary range of tasks may not be obvious to the humans designing a meta-learner, in which case the meta-learner should have the flexibility to learn the best way to solve the tasks it is presented with. Such a meta-learner would need to have an expressive, versatile model architecture, in order to learn a range of strategies in a variety of domains.

Meta-learning can be formalized as a sequence-to-sequence problem; in existing approaches that adopt this view, the bottleneck is in the meta-learner's ability to internalize and refer to past experience. Thus, we propose a class of model architectures that addresses this shortcoming: we combine temporal convolutions, which enable the meta-learner to aggregate contextual information from past experience, with causal attention, which allow it to pinpoint specific pieces of information within that context. We evaluate this Simple Neural AttenIve Learner (SNAIL) on several heavily-benchmarked meta-learning tasks, including the Omniglot and mini-Imagenet datasets in supervised learning, and multi-armed bandits, tabular Markov Decision processes (MDPs), visual navigation, and continuous control in reinforcement learning. In all domains, SNAIL achieves state-of-the-art performance by significant margins, outperforming methods that are domain-specific or rely on built-in algorithmic priors.

---

[*]Authors contributed equally and are listed in alphabetical order.
[†]Part of this work was done at OpenAI.

## 2 META-LEARNING PRELIMINARIES

Before we describe SNAIL in detail, we will introduce notation and formalize the meta-learning problem. As briefly discussed in Section 1, the goal of meta-learning is generalization across tasks rather than across data points. Each task $\mathcal{T}_i$ is episodic and defined by inputs $x_t$, outputs $a_t$, a loss function $\mathcal{L}_i(x_t, a_t)$, a transition distribution $P_i(x_t|x_{t-1}, a_{t-1})$, and an episode length $H_i$. A meta-learner (with parameters $\theta$) models the distribution $\pi(a_t|x_1, \ldots, x_t; \theta)$. Given a distribution over tasks $\mathcal{T} = P(\mathcal{T}_i)$, the meta-learner's objective is to minimize its expected loss with respect to $\theta$.

$$\min_{\theta} \mathbb{E}_{\mathcal{T}_i \sim \mathcal{T}} \left[ \sum_{t=0}^{H_i} \mathcal{L}_i(x_t, a_t) \right],$$
$$\text{where } x_t \sim P_i(x_t|x_{t-1}, a_{t-1}), \ a_t \sim \pi(a_t|x_1, \ldots, x_t; \theta)$$

A meta-learner is trained by optimizing this expected loss over tasks (or mini-batches of tasks) sampled from $\mathcal{T}$. During testing, the meta-learner is evaluated on unseen tasks from a different task distribution $\widetilde{\mathcal{T}} = P(\widetilde{\mathcal{T}}_i)$ that is similar to the training task distribution $\mathcal{T}$.

## 3 A SIMPLE NEURAL ATTENTIVE LEARNER

The key principle motivating our approach is simplicity and versatility: a meta-learner should be universally applicable to domains in both supervised and reinforcement learning. It should be generic and expressive enough to learn an optimal strategy, rather than having the strategy already built-in.

Santoro et al. (2016) considered a similar formulation of the meta-learning problem, and explored using recurrent neural networks (RNNs) to implement a meta-learner. Although simple and generic, their approach is significantly outperformed by methods that are hand-designed to exploit domain or algorithmic knowledge (methods which we survey in Section 4). We hypothesize that this is because traditional RNN architectures propagate information by keeping it in their hidden state from one timestep to the next; this temporally-linear dependency bottlenecks their capacity to perform sophisticated computation on a stream of inputs.

van den Oord et al. (2016a) introduced a class of architectures that generate sequential data (in their case, audio) by performing dilated 1D-convolutions over the temporal dimension. These temporal convolutions (TC) are causal, so that the generated values at the next timestep are only influenced by past timesteps and not future ones. Compared to traditional RNNs, they offer more direct, high-bandwidth access to past information, allowing them to perform more sophisticated computation over a temporal context of fixed size. However, to scale to long sequences, the dilation rates generally increase exponentially, so that the required number of layers scales logarithmically with the sequence length. Hence, they have coarser access to inputs that are further back in time; their bounded capacity and positional dependence can be undesirable in a meta-learner, which should be able to fully utilize increasingly large amounts of experience.

In contrast, soft attention (in particular, the style used by Vaswani et al. (2017a)) allows a model to pinpoint a specific piece of information from a potentially infinitely-large context. It treats the context as an unordered key-value store which it can query based on the content of each element. However, the lack of positional dependence can also be undesirable, especially in reinforcement learning, where the observations, actions, and rewards are intrinsically sequential.

Despite their individual shortcomings, temporal convolutions and attention complement each other: while the former provide high-bandwidth access at the expense of finite context size, the latter provide pinpoint access over an infinitely large context. Hence, we construct SNAIL by combining the two: we use temporal convolutions to produce the context over which we use a causal attention operation. By interleaving TC layers with causal attention layers, SNAIL can have high-bandwidth access over its past experience *without* constraints on the amount of experience it can effectively use. By using attention at multiple stages within a model that is trained end-to-end, SNAIL can learn what pieces of information to pick out from the experience it gathers, as well as a feature representation that is amenable to doing so easily. As an additional benefit, SNAIL architectures are easier to train than traditional RNNs such as LSTM or GRUs (where the underlying optimization can be difficult because of the temporally-linear hidden state dependency) and can be efficiently implemented so that an entire sequence can be processed in a single forward pass. Figure 1 provides an illustration of SNAIL, and we discuss architectural components in Section 3.1.

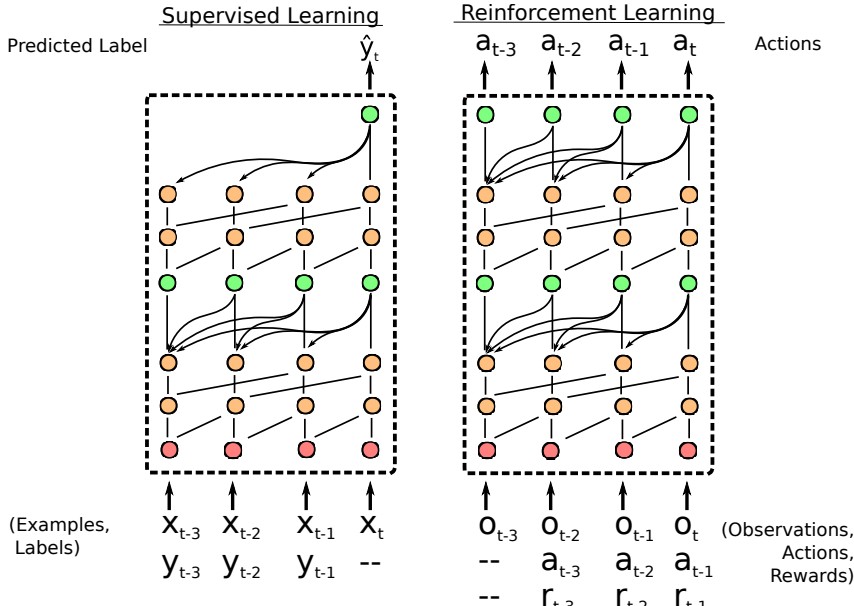

Figure 1: Overview of our simple neural attentive learner (SNAIL); in this example, two blocks of TC layers (orange) are interleaved with two causal attention layers (green). The same class of model architectures can be applied to both supervised and reinforcement learning.

In supervised settings, SNAIL receives as input a sequence of example-label pairs $(x_1, y_1), \ldots, (x_{t-1}, y_{t-1})$ for timesteps $1, \ldots, t-1$, followed by an unlabeled example $(x_t, -)$. It then outputs its prediction for $x_t$ based on the previous labeled examples it has seen.

In reinforcement-learning settings, it receives a sequence of observation-action-reward tuples $(o_1, -, -), \ldots, (o_t, a_{t-1}, r_{t-1})$. At each time $t$, it outputs a distribution over actions $a_t$ based on the current observation $o_t$ as well as previous observations, actions, and rewards. Crucially, following existing work in meta-RL (Duan et al., 2016; Wang et al., 2016), we preserve the internal state of a SNAIL across episode boundaries, which allows it to have memory that spans multiple episodes. The observations also contain a binary input that indicates episode termination.

## 3.1 MODULAR BUILDING BLOCKS

We compose SNAIL architectures using a few primary building blocks. Below, we provide pseudocode for applying each block to a matrix ("inputs" in the pseudocode) of size (sequence length) × (input dimensionality). Note that, if any of the inputs are images, we employ an additional (spatial) convolutional network that converts the image into a feature vector before it is passed into the SNAIL. Figure 2 illustrates the different blocks visually.

Many techniques have been proposed to increase the capacity or accelerate the training of deep convolutional architectures, including batch normalization (Ioffe & Szegedy (2015)), residual connections (He et al. (2016)), and dense connections (Huang et al. (2016)). We found that these techniques greatly improved the expressive capacity and training speed of SNAILs, but that no particular choice of residual/dense configurations was essential for good performance (we explore the robustness of SNAILs to architectural choices in Appendix B).

A *dense block* applies a single causal 1D-convolution with dilation rate $R$ and $D$ filters (we used kernel size 2 in all experiments), and then concatenates the result with its input. We used the gated activation function (line 3) introduced by van den Oord et al. (2016a;b).

```
1: function DENSEBLOCK(inputs, dilation rate R, number of filters D):
2:     xf, xg = CausalConv(inputs, R, D), CausalConv(inputs, R, D)
3:     activations = tanh(xf) * sigmoid(xg)
4:     return concat(inputs, activations)
```

A *TC block* consists of a series of dense blocks whose dilation rates increase exponentially until their receptive field exceeds the desired sequence length:

---
1: **function** TCBLOCK(inputs, sequence length $T$, number of filters $D$):
2:     **for** $i$ in $1, \ldots, \lceil \log_2 T \rceil$ **do**
3:         inputs = DenseBlock(inputs, $2^i$, $D$)
4:     **return** inputs

---

A *attention block* performs a single key-value lookup; we style this operation after the self-attention mechanism proposed by Vaswani et al. (2017a):

---
1: **function** ATTENTIONBLOCK(inputs, key size $K$, value size $V$):
2:     keys, query = affine(inputs, $K$), affine(inputs, $K$)
3:     logits = matmul(query, transpose(keys))
4:     probs = CausallyMaskedSoftmax(logits / $\sqrt{K}$)
5:     values = affine(inputs, $V$)
6:     read = matmul(probs, values)
7:     **return** concat(inputs, read)

---

where CausallyMaskedSoftmax($\cdot$) zeros out the appropriate probabilities before normalization, so that a particular timestep's query cannot have access to future keys/values.

(a) Dense Block (dilation rate R, D filters)

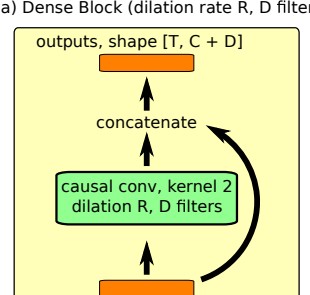

(b) Attention Block (key size K, value size V)

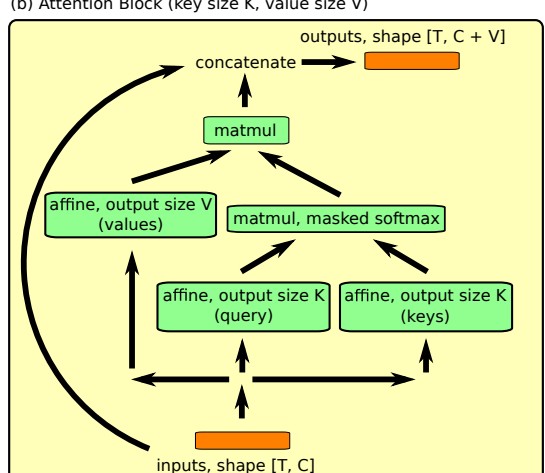

Figure 2: Two of the building blocks that compose SNAIL architectures. (a) A dense block applies a causal 1D-convolution, and then concatenates the output to its input. A TC block (not pictured) applies a series of dense blocks with exponentially-increasing dilation rates. (b) A attention block performs a (causal) key-value lookup, and also concatenates the output to the input.

## 4 RELATED WORK

Pioneered by Schmidhuber (1987); Naik & Mammone (1992); Thrun & Pratt (1998), meta-learning is not a new idea. A key tradeoff central to many recent meta-learning approaches is between performance and generality; we discuss several notable methods and how they fit into this paradigm.

Graves et al. (2014) investigated the use of recurrent neural networks (RNNs) to solve algorithmic tasks. They experimented with a meta-learner implemented by an LSTM, but their results suggested that LSTM architectures are ill-equipped for these kinds of tasks. They then designed a more sophisticated RNN architecture, where an LSTM controller was coupled to an external memory bank from which it can read and write, and demonstrated that these memory-augmented neural networks (MANNs) achieved substantially better performance than LSTMs. Santoro et al. (2016) evaluated both LSTM and MANN meta-learners on few-shot image classification, and confirm the inadequacy

of the LSTM architecture. These approaches are generic, but MANNs feature a complicated memory-addressing architecture that is difficult to train – they still suffer from the same temporally-linear hidden-state dependencies as LSTMs.

In response, several approaches have demonstrated good performance in few-shot classification with specialized neural network architectures. Koch (2015) used a Siamese network that was trained to predict whether two images belong to the same class. Vinyals et al. (2016) learned an embedding function and used cosine distance in an attention kernel to judge image similarity. Snell et al. (2017) employed a similar approach to Vinyals et al. (2016), based on Euclidean distance metrics. All three methods work well within the context of classification, but are not readily applicable to other domains, such as reinforcement learning. They perform well because their architectures have been designed to exploit domain knowledge, but ideally we would like a meta-learner that is not constrained to a particular problem type.

A number of methods consider a meta-learner that makes updates to the parameters of a traditional learner (Bengio et al., 1992; Hochreiter et al., 2001). Andrychowicz et al. (2016) and Li & Malik (2017) investigated the setting of learning to optimize, where the learner is an objective function to minimize, and the meta-learner uses the gradients of the learner to perform the optimization. Their meta-learner was implemented by an LSTM and the strategy that it learned can be interpreted as a gradient-based optimization algorithm; however, it is unclear whether the learned optimizers are substantially better than existing SGD-based methods.

Ravi & Larochelle (2017) extended this idea, using a similar LSTM meta-learner in a few-shot classification setting, where the traditional learner was a convolutional-network-based classifier. In this setting, the meta-learning algorithm is decomposed into two parts: the traditional learner's initial parameters are trained to be suitable for fast gradient-based adaptation; the LSTM meta-learner is trained to be an optimization algorithm adapted for meta-learning tasks. Finn et al. (2017) explored a special case where the meta-learner is constrained to use ordinary gradient descent to update the learner and showed that this simplified model (known as MAML) can achieve equivalent performance. Munkhdalai & Yu (2017) explored a more sophisticated weight update scheme that yielded minor performance improvements on few-shot classification.

All of the methods discussed in the previous paragraph have the benefit of being domain independent, but they explicitly encode a particular strategy for the meta-learner to follow (namely, adaptation via gradient descent at test time). In a particular domain, there may exist better strategies that exploit the structure of the task, but gradient-based methods will be unable to discover them. In contrast, SNAIL presents an alternative paradigm where a generic architecture has the capacity to learn an algorithm that exploits domain-specific task structure.

Duan et al. (2016) and Wang et al. (2016) both investigated meta-learning in reinforcement-learning domains using traditional RNN architectures (GRUs and LSTMs). In addition, Finn et al. (2017) experimented with fast adaptation of policies in continuous control, where the meta-learner was trained on a distribution of closely-related locomotion tasks. In Section 5.2, we benchmark SNAIL against MAML and an LSTM-based meta-learner on the tasks considered by these works.

## 5 EXPERIMENTS

Our experiments were designed to investigate the following questions:

- How does SNAIL's generality affect its performance on a range of meta-learning tasks?
- How does its performance compare to existing approaches that are specialized to a particular task domain, or have elements of a high-level strategy already built-in?
- How does SNAIL scale with high-dimensional inputs and long-term temporal dependencies?

### 5.1 FEW-SHOT IMAGE CLASSIFICATION

In the few-shot classification setting, we wish to classify data points into $N$ classes when we only have a small number $(K)$ of labeled examples per class. A meta-learner is readily applicable, because it learns how to compare input points, rather than memorize a specific mapping from points to classes.

The Omniglot and mini-ImageNet datasets for few-shot image classification are the standard benchmarks in supervised meta-learning. Introduced by Lake et al. (2011), Omniglot consists of black-

and-white images of handwritten characters gathered from 50 languages, for a total of 1632 different classes with 20 instances per class. Like prior works, we downsampled the images to $28 \times 28$ and randomly selected 1200 classes for training and 432 for testing. We performed the same data augmentation proposed by Santoro et al. (2016), forming new classes by rotating each member of an existing class by a multiple of 90 degrees.

Mini-ImageNet is a more difficult benchmark; a subset of the well-known ImageNet dataset, it consists of $84 \times 84$ color images from 100 different classes with 600 instances per class. We used the split released by Ravi & Larochelle (2017) and used by a number of other works, with 64 classes for training, 16 for validation, and 20 for testing.

To evaluate a SNAIL on the $N$-way, $K$-shot problem, we sample $N$ classes from the overall dataset and $K$ examples of each class. We then feed the corresponding $NK$ example-label pairs to the SNAIL in a random order, followed by a new, unlabeled example from one of the $N$ classes. We report the average accuracy on this last, $(NK + 1)$-th timestep.

We tested SNAIL on 5-way Omniglot, 20-way Omniglot, and 5-way mini-ImageNet. For each of these three splits, we trained the SNAIL on episodes where the number of shots $K$ was chosen uniformly at random from 1 to 5 (*note that this is unlike prior works, who train separate models for each shot*). For a $K$-shot episode within an $N$-way problem, the loss was simply the average cross-entropy between the predicted and true label on the $(NK + 1)$-th timestep. We train both the SNAIL and the feature-extracting embedding network in an end-to-end fashion using Adam (Kingma & Ba, 2015) For a complete description of the specifics SNAIL and embedding architectures we used, we refer the reader to Appendix A.

Table 1 displays our results on 5-way and 20-way Omniglot, and Table 2 respectively for 5-way mini-ImageNet. We see that SNAIL outperforms state-of-the-art methods that are extensively hand-designed, and/or domain-specific. It significantly exceeds the performance of methods such as Santoro et al. (2016) that are similarly simple and generic. In Appendix B, we conduct a number of ablations to analyse SNAIL's performance.

Table 1: 5-way and 20-way, 1-shot and 5-shot classification accuracies on Omniglot, with 95% confidence intervals where available. For each task, the best-performing method is highlighted, along with any others whose confidence intervals overlap.

| Method | 5-Way Omniglot | | 20-Way Omniglot | |
|---|---|---|---|---|
| | 1-shot | 5-shot | 1-shot | 5-shot |
| Santoro et al. (2016) | 82.8% | 94.9% | – | – |
| Koch (2015) | 97.3% | 98.4% | 88.2% | 97.0% |
| Vinyals et al. (2016) | 98.1% | 98.9% | 93.8% | 98.5% |
| Finn et al. (2017) | **98.7% $\pm$ 0.4%** | **99.9% $\pm$ 0.3%** | 95.8% $\pm$ 0.3% | 98.9% $\pm$ 0.2% |
| Snell et al. (2017) | 97.4% | 99.3% | 96.0% | 98.9% |
| Munkhdalai & Yu (2017) | **98.9%** | – | 97.0% | – |
| SNAIL, Ours | **99.07% $\pm$ 0.16%** | **99.78% $\pm$ 0.09%** | **97.64% $\pm$ 0.30%** | **99.36% $\pm$ 0.18%** |

Table 2: 5-way, 1-shot and 5-shot classification accuracies on mini-ImageNet, with 95% confidence intervals where available. For each task, the best-performing method is highlighted, along with any others whose confidence intervals overlap.

| Method | 5-Way Mini-ImageNet | |
|---|---|---|
| | 1-shot | 5-shot |
| Vinyals et al. (2016) | 43.6% | 55.3% |
| Finn et al. (2017) | 48.7% $\pm$ 1.84% | 63.1% $\pm$ 0.92% |
| Ravi & Larochelle (2017) | 43.4% $\pm$ 0.77% | 60.2% $\pm$ 0.71% |
| Snell et al. (2017) | 46.61% $\pm$ 0.78% | 65.77% $\pm$ 0.70% |
| Munkhdalai & Yu (2017) | 49.21% $\pm$ 0.96% | – |
| SNAIL, Ours | **55.71% $\pm$ 0.99%** | **68.88% $\pm$ 0.92%** |

## 5.2 REINFORCEMENT LEARNING

Reinforcement learning features a number of challenges that supervised learning does not, including long-term temporal dependencies (as the experienced states and rewards may depend on actions taken many timesteps ago) as well as balancing exploration and exploitation. To explore SNAIL's ability to learn RL algorithms, we evaluate it on four different domains from prior work in meta-RL[1]:

- Multi-armed bandits (Duan et al., 2016; Wang et al., 2016): the agent interacts with a set of arms whose reward distributions are unknown. Although its actions do not affect its state, exploration and exploitation are both essential: an optimal agent must initially explore by sampling different arms, but later exploit its knowledge by repeatedly selecting the best arm.
- Tabular MDPs (Duan et al., 2016; Wang et al., 2016): we procedurally generate random MDPs and allow the agent to act within each one for multiple episodes. Since every MDP is different, a meta-learner cannot simply memorize the ones it is trained on; it must actually learn an algorithm for solving MDPs.
- Visual navigation (Duan et al., 2016; Wang et al., 2016): the agent must navigate randomly-generated mazes to find a randomly-located goal, using only visual observations as input. It is allowed to interact with the same maze/goal configuration for two episodes, so an optimal agent should explore the maze on the first episode to find the goal, and then go directly to the goal on the second episode. This task features many of the common challenges in deep RL, including high-dimensional observations, partial observability, and sparse rewards.
- Continuous control (Finn et al., 2017): we consider a suite of simulated locomotion tasks. Although the environment dynamics are complex, the underlying task distribution is quite narrow. As a result, there is significant task structure for a meta-learner to exploit; the optimal strategy is closer to task-identification than a true RL algorithm.

On each of these domains, we trained a SNAIL, along with two meta-learning baselines:

- An LSTM-based meta-learner, as concurrently proposed by Duan et al. (2016); Wang et al. (2016). We refer to this method as "LSTM" in the tables and figures in subsequent sections.
- MAML, the method introduced by Finn et al. (2017). It trains the initial parameters of a policy to achieve maximal performance after one (policy) gradient update on a new task.

We also conducted some ablation experiments, which are detailed in Appendix D.

In all domains, we trained the meta-learners using trust region policy optimization with generalized advantage estimation (TRPO with GAE; Schulman et al. (2015; 2016)); the SNAIL architectures and TRPO/GAE hyperparameters are detailed in Appendix C.

In the bandit and MDP domains, there exist a number of human-designed algorithms with various optimality guarantees (which we discuss in more depth in the subsequent sections). Although there isn't much task structure for a meta-learner to exploit, the existence of upper bounds on asymptotic performance let us evaluate the optimality of a meta-learned algorithm.

However, the true utility of a meta-learner is that it can learn an algorithm specialized to the particular distribution of tasks it is trained on. We evaluate this in the visual navigation and continuous control domains, where there is significant task structure for the meta-learner to exploit, but no optimal algorithms are known to exist due to the task complexity.

### 5.2.1 MULTI-ARMED BANDITS

In our bandit experiments (styled after Duan et al. (2016)), each of $K$ arms gives rewards according to a Bernoulli distribution whose parameter $p \in [0, 1]$ is chosen randomly at the start of each episode of length $N$. At each timestep, the meta-learner receives previous timestep's reward, along with a one-hot encoding of the corresponding arm selected. It outputs a discrete probability distribution over the $K$ arms; the selected arm is determined by sampling from this distribution.

As an oracle, we consider the Gittins index (Gittins, 1979), the Bayes optimal solution in the discounted, infinite horizon setting. Since it is only optimal as $N \to \infty$, a meta-learner can outperform it for smaller $N$ by choosing to exploit sooner.

Following Duan et al. (2016), we tested all combinations of $N = 10, 100, 500$ and $K = 5, 10, 50$. We also tested the additional case of $N = 1000, K = 50$ to further evaluate the scalability of SNAIL to longer sequences. We report the mean reward per episode for each setting; the results are given in Table 3 with 95% confidence intervals where available. We found that training MAML was too computationally expensive for $N = 500, 1000$; hence we omit those results from Table 3.

---

[1] Some video results can be found at https://sites.google.com/view/snail-iclr-2018/.

Table 3: Results on multi-arm bandit problems. For each, we highlighted the best performing method, and any others whose performance is not statistically-significantly different (based on a one-sided $t$-test with $p = 0.05$). Except for SNAIL and MAML, we report the results from Duan et al. (2016).

| Setup $(N, K)$ | Gittins (optimal as $N \to \infty$) | Method | | | |
|---|---|---|---|---|---|
| | | Random | LSTM | MAML | SNAIL (ours) |
| $10, 5$ | **6.6** | 5.0 | **6.7** | $6.5 \pm 0.1$ | $\mathbf{6.6 \pm 0.1}$ |
| $10, 10$ | **6.6** | 5.0 | **6.7** | $\mathbf{6.6 \pm 0.1}$ | $\mathbf{6.7 \pm 0.1}$ |
| $10, 50$ | 6.5 | 5.1 | **6.8** | $\mathbf{6.6 \pm 0.1}$ | $\mathbf{6.7 \pm 0.1}$ |
| $100, 5$ | **78.3** | 49.9 | **78.7** | $67.1 \pm 1.1$ | $\mathbf{79.1 \pm 1.0}$ |
| $100, 10$ | **82.8** | 49.9 | **83.5** | $70.1 \pm 0.6$ | $\mathbf{83.5 \pm 0.8}$ |
| $100, 50$ | **85.2** | 49.8 | **84.9** | $70.3 \pm 0.4$ | $\mathbf{85.1 \pm 0.6}$ |
| $500, 5$ | **405.8** | 249.8 | 401.5 | – | $\mathbf{408.1 \pm 4.9}$ |
| $500, 10$ | **437.8** | 249.0 | **432.5** | – | $\mathbf{432.4 \pm 3.5}$ |
| $500, 50$ | **463.7** | 249.6 | 438.9 | – | $442.6 \pm 2.5$ |
| $1000, 50$ | **944.1** | 499.8 | 847.43 | – | $889.8 \pm 5.6$ |

### 5.2.2 TABULAR MDPS

In our tabular MDP experiments (also following Duan et al. (2016)), each MDP had 10 states and 5 actions (both discrete); the reward for each (state, action)-pair followed a normal distribution with unit variance where the mean was sampled from $\mathcal{N}(1, 1)$, and the transitions are sampled from a flat Dirichlet distribution (the latter is a commonly used prior in Bayesian RL) with random parameters. We allowed each meta-learner to interact with an MDP for $N$ episodes of length 10. As input, they received one-hot encodings of the current state and previous action, the previous reward received, and a binary flag indicating termination of the current episode.

In addition to a random agent, we consider the follow human-designed algorithms as baselines.

- PSRL (Strens, 2000): a Bayesian method that estimate the belief over the current MDP parameters. At the start of each of the $N$ episodes, it samples an MDP from the current posterior, and acts according to the optimal policy for the rest of the episode.
- OPSRL (Osband & Van Roy, 2017): an optimistic variant of PSRL.
- UCRL2 (Jaksch et al., 2010): uses an extended value iteration procedure to compute an optimistic MDP under the current belief.
- $\epsilon$-greedy: with probability $1 - \epsilon$, act optimally against the MAP estimate according to the current posterior (which is updated once per episode).

As an oracle, we run value iteration for 10 iterations (the episode length) on each MDP. Value iteration is optimal when the MDP parameters (reward function, transition probabilities) are known; thus, the resulting values provide an upper bound on the performance of any algorithm, whether human-designed or meta-learned (which do not receive the MDP parameters).

We tested $N = 10, 25, 50, 75, 100$; in Table 4, we report the performance normalized by the value-iteration upper bound. As $N$ increases, performance should approach 1, as the algorithm learns more about the current MDP. Similarly to the bandit experiments, we could not train MAML successfully for $N = 50, 75, 100$. In Figure 3, we show learning curves of SNAIL and LSTM.

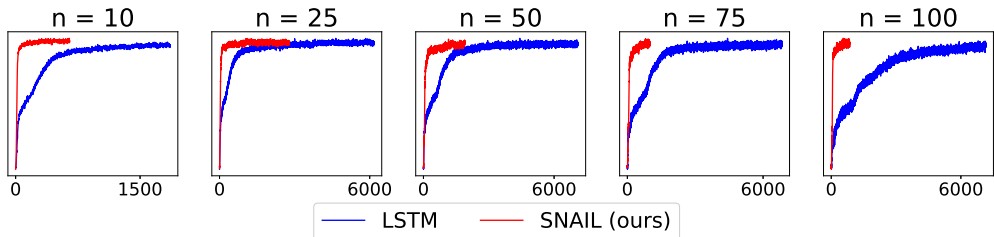

Figure 3: Learning curves of SNAIL (red) and LSTM (blue) on the random MDP task for different values of $N$. The horizontal axis is the TRPO iteration, and the vertical is average reward.

Table 4: Performance on tabular MDPs, scaled by the average reward achieved by value iteration. As before, we highlight the best-performing method, and any others whose performance is not statistically-significantly different (using the same one-sided $t$-test with $p = 0.05$). Except for SNAIL and MAML, we report the values from Duan et al. (2016).

| $N$ | Method | | | | | | | |
|---|---|---|---|---|---|---|---|---|
| | Random | $\epsilon$-greedy | PSRL | OPSRL | UCRL2 | LSTM | MAML | SNAIL (ours) |
| 10 | 0.482 | 0.640 | 0.665 | 0.694 | 0.706 | 0.752 | 0.563 | **0.766 $\pm$ 0.001** |
| 25 | 0.482 | 0.727 | 0.788 | 0.819 | 0.817 | 0.859 | 0.591 | **0.862 $\pm$ 0.001** |
| 50 | 0.481 | 0.793 | 0.871 | 0.897 | 0.885 | 0.902 | – | **0.908 $\pm$ 0.003** |
| 75 | 0.482 | 0.831 | 0.910 | **0.931** | 0.917 | 0.918 | – | **0.930 $\pm$ 0.002** |
| 100 | 0.481 | 0.857 | 0.934 | **0.951** | 0.936 | 0.922 | – | 0.941 $\pm$ 0.003 |

### 5.2.3 CONTINUOUS CONTROL

We consider the set of tasks introduced by Finn et al. (2017), in which two simulated robots (a planar cheetah and a 3D-quadruped ant) have to run in a particular direction or at a specified velocity (the direction or velocity are chosen randomly and not told to the agent). In the goal direction experiments, the reward is the magnitude of the robot's velocity in either the forward or backward direction, and in the goal velocity experiments, the reward is the negative absolute value between its current forward velocity and the goal. The observations are the robot's joint angles and velocities, and the actions are its joint torques. For each of these four task distributions ({ant, cheetah} $\times$ {goal velocity, goal direction}), Finn et al. (2017) trained a policy to maximize its performance after one policy gradient update using 20 episodes (40 for ant), of 200 timesteps each, on a newly sampled task.

We trained both SNAIL and LSTM on each of these four task categories. Since they do not update their parameters at test time (instead incorporating experience through their hidden state), SNAIL and LSTM receive as input the previous action, previous reward, and an episode-termination flag in addition to the current observation. We found that two episodes of interaction was sufficient for these meta-learners to adapt to a task, and that unrolling them for longer did not improve performance.

In Figure 4, we show how the different methods adapt to a new task. As an oracle, we sampled tasks from each distribution, and trained a separate policy for each task. We plot the average performance of the oracle policies for each task distribution as an upper bound on a meta-learner's performance.

Qualitatively, we can think of MAML as applying a general-purpose strategy (namely, gradient descent) to a distribution of highly-structured tasks. In contrast, SNAIL and LSTM are able to specialize themselves based on the shared task structure, enabling them to identify the task within the initial timesteps of the first episode, and then act optimally thereafter.

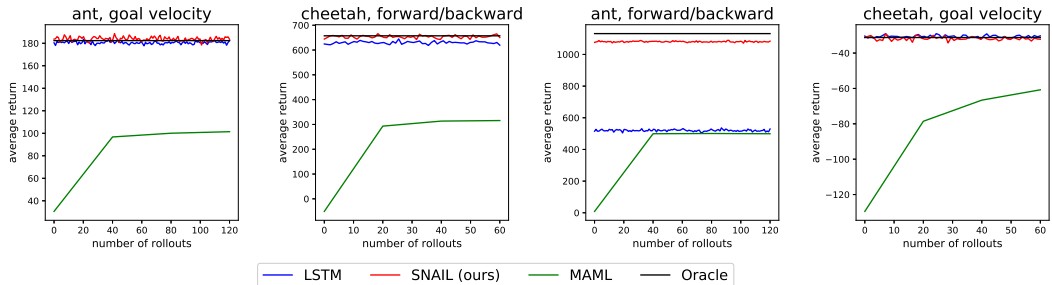

Figure 4: Test-time adaptation curves on simulated locomotion tasks for SNAIL, LSTM, and MAML (which was unrolled for three policy gradient updates). Since SNAIL incorporates experience through its hidden state, it can exploit common task structure to perform optimally within a few timesteps.

### 5.2.4 VISUAL NAVIGATION

Both Duan et al. (2016) and Wang et al. (2016) consider the task of visual navigation, where the agent must find a target in a maze using only visual inputs. The former used randomly-generated mazes and target positions, while the latter used a fixed maze and only four different target positions. Hence, we evaluated SNAIL on the former, more challenging task. The observations the agent receives are

$30 \times 40$ first-person images, and the actions it can take are {step forward, turn slightly left, turn slightly right}. We constructed a training dataset and two test datasets (unseen mazes of the same and larger size, respectively), each with 1000 mazes. The agents were allowed to interact with each maze for 2 episodes, with episode length 250 (1000 in the larger mazes). The starting and goal locations were chosen randomly for each trial but remained fixed within each pair of episodes. The agents received rewards of +1 for reaching the target (which resulted in the episode terminating), -0.01 at each timestep, to encourage it to reach the goal faster, and -0.001 for hitting the wall. Figure 5 depicts an example of the observations as well as sample maze layouts.

We evaluate each method using the average episode length, for both the first and second episode within a trial. The results are displayed in Table 5. Since MAML scaled poorly to long sequences in the bandit and MDP domains, we did not evaluate it on this domain; the computational expense was prohibitively high. Qualitatively, we observe that the optimal strategy does indeed emerge: the SNAIL agent explores the maze during the first episode, and then, after finding the goal, goes directly there on the second episode (the LSTM agent also exhibits this behavior, but has a harder time remembering where the goal is). An illustration is depicted in Figure 5.

Table 5: Average time to find the goal on each episode in the small and large mazes. SNAIL solves the mazes the fastest, and improves the most from the first to second episode.

| Method | Small Maze | | Large Maze | |
|---|---|---|---|---|
| | Episode 1 | Episode 2 | Episode 1 | Episode 2 |
| Random | $188.6 \pm 3.5$ | $187.7 \pm 3.5$ | $420.2 \pm 1.2$ | $420.8 \pm 1.2$ |
| LSTM | $52.4 \pm 1.3$ | $39.1 \pm 0.9$ | $180.1 \pm 6.0$ | $150.6 \pm 5.9$ |
| SNAIL (ours) | $\mathbf{50.3 \pm 0.3}$ | $\mathbf{34.8 \pm 0.2}$ | $\mathbf{140.5 \pm 4.2}$ | $\mathbf{105.9 \pm 2.4}$ |

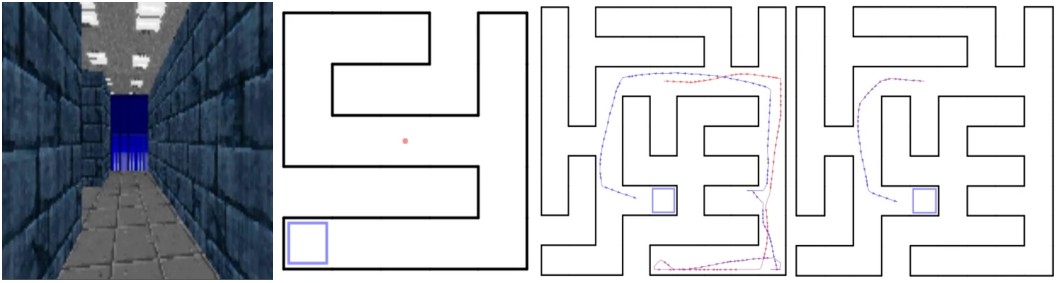

Figure 5: From left to right: (a) A (higher-resolution) example of the observations the agent receives. (b) An example of the mazes used for training (goal shown in blue). (c) The movement of the SNAIL on its first episode in a larger maze, exploring the maze until it finds the goal. (d) The SNAIL's path during its second episode in the same maze as (c). Remembering the goal location, it navigates there directly on the second episode. Maps like in (b), (c), (d) are used for visualization but not available to the agent. In (c), (d), the color progression from red to blue indicates the passage of time (red earlier).

## 6 CONCLUSION AND FUTURE WORK

We presented a simple and generic class of architectures for meta-learning, motivated by the need for a meta-learner to quickly incorporate and refer to past experience. Our simple neural attentive learner (SNAIL) utilizes a novel combination of temporal convolutions and causal attention, two building blocks of sequence-to-sequence models that have complementary strengths and weaknesses. We demonstrate that SNAIL achieves state-of-the-art performance by significant margins on all of the most-widely benchmarked meta-learning tasks in both supervised and reinforcement learning, without relying on any application-specific architectural components or algorithmic priors.

Although we designed SNAIL with meta-learning in mind, it would likely excel at other sequence-to-sequence tasks, such as language modeling or translation; we plan to explore this in future work.

Another interesting idea would be to train an meta-learner that can attend over its entire lifetime of experience (rather than only a few recent episodes, as in this work). An agent with this lifelong memory could learn faster and generalize better; however, to keep the computational requirements practical, it would also need to learn how to decide what experiences are worth remembering.

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

APPENDIX

## A    FEW-SHOT CLASSIFICATION ARCHITECTURES

With the building blocks defined in Section 3.1, we can concisely describe SNAIL architectures. We used the same SNAIL architecture for both Omniglot and mini-Imagenet. For the $N$-way, $K$-shot problem, the sequence length is $T = NK + 1$, and we used the following: AttentionBlock(64, 32), TCBlock($T$, 128), AttentionBlock(256, 128), TCBlock($T$, 128), AttentionBlock(512, 256), followed by a final $1 \times 1$ convolution with $N$ filters.

For the Omniglot dataset, we used the same embedding network architecture as all prior works, which repeat the following block four times { 3x3 conv (64 channels), batch norm, ReLU, 2x2 max pool }, and then apply a single fully-connected layer to output a 64-dimensional feature vector.

For mini-Imagenet, existing gradient-descent-based methods (Ravi & Larochelle, 2017; Finn et al., 2017), which update their model's weights during testing, used the same network structure as the Omniglot network but reduced the number of channels to 32, in spite of the significantly-increased complexity of the images. We found that this shallow embedding network did not make adequate use of SNAIL's expressive capacity, and opted to to use a deeper embedding network to prevent underfitting (in Appendix B, we conduct ablations regarding this decision). Illustrated in Figure 6, our embedding was a smaller version of the ResNet (He et al., 2016) architectures commonly used for the full Imagenet dataset.

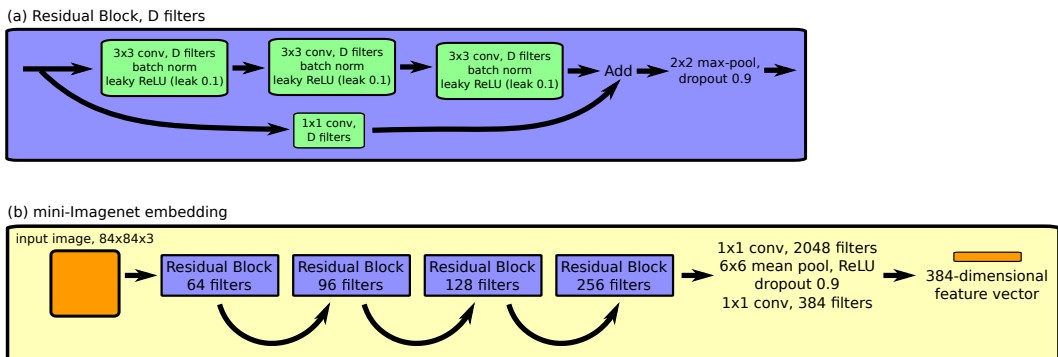

Figure 6: (a) A residual block within our mini-Imagenet embedding. (b) The embedding, a smaller version of ResNet (He et al., 2016), uses several of the residual blocks depicted in (a).

## B   FEW-SHOT CLASSIFICATION: ABLATIONS

To investigate the contribution of different components (TC, attention, and deeper embedding in the case of mini-Imagenet) to SNAIL's performance, we conducted a number of ablations, which are summarized in Table 6. From these ablations, we draw two conclusions:

- Both TC and attention layers are essential for maximal performance. When we remove either one, the resulting model is still competitive with other state-of-the-art methods, but the combination yields the best performance. Notably, compared to the full model, using only TC layers results in similar 1-shot performance but worse 5-shot. In Section 3 we discussed how temporal convolutions have coarser access to inputs farther back in time; this illustrates that this effect is relevant even at sequence length 26.

- SNAIL's improved performance is not purely a result of the deeper embedding network. Gradient-based methods (we tested MAML; Finn et al. (2017)) overfit significantly when they use our embedding, and domain-specific RNN-based methods (Vinyals et al., 2016) don't utilize the extra capacity as well as SNAIL does.

Table 6: The ablations we conducted on the few-shot classification task. From these, we conclude that (i) both TC and attention are essential for the best performance, and (ii) SNAIL's improved performance cannot be entirely explained by the deeper embedding.

| Ablation | Result |
|---|---|
| Replace SNAIL with stacked LSTM. Varied number of layers and their sizes (with similar number of parameters to SNAIL). | 5-way Omniglot: $78.1\%$ and $90.8\%$ (1-shot, 5-shot). We were unable to successfully train this method on mini-Imagenet. |
| SNAIL with shallow mini-Imagenet embedding. | 5-way mini-Imagenet: $45.1\%$ and $55.2\%$ (1-shot, 5-shot). |
| MAML (Finn et al., 2017), a state-of-the-art gradient-based method, with our deeper mini-Imagenet embedding. | It overfits tremendously; for 1-shot, 5-way mini-ImageNet: $30.1\%$ & $75.2\%$ on the test and training set respectively. MAML trains separate models for 1-shot and 5-shot; we didn't train a 5-shot model because the 1-shot did so poorly. |
| SNAIl, no TC layers (only attention). This is a generalization of the method used by Vinyals et al. (2016), as they only use a single attentive read and explicitly force the keys to be features of the image and the values to be the labels. We experimented with multiple parallel reads (often referred to as multiple heads) as well as up to three consecutive attentive blocks. | On 5-way and 20-way Omniglot: equivalent performance to the full model. 5-way mini-Imagenet: 49.9% and 63.9% (1-shot, 5-shot). |
| SNAIL, no attention (TC layers only). | On 5-way Omniglot: $98.8\%$ and $99.2\%$ (1-shot, 5-shot). We were unable to train 20-way Omniglot using this method. On 5-way mini-Imagenet: $55.1\%$ and $61.2\%$. |

In an attempt to analyse the learned feature representation, we tried using the features learned by the Omniglot embedding in a nearest-neighbor classifier, using both cosine and Euclidean distance. On 5-way Omniglot, this achieves $65.1\%$ and $67.1\%$ (1-shot and 5-shot) for Euclidean distance and $67.7\%$ and $68.3\%$ for cosine. Although SNAIL must be comparing images in order to successfully make few-shot predictions, this suggests that the strategy it learns is more sophisticated than either of these distance metrics. We contrast this with Vinyals et al. (2016) and Snell et al. (2017), who explicitly enforce such representations on the meta-learned strategy.

In addition, we investigated how sensitive SNAILs are to architectural design choices by sampling random permutations of the different components introduced in Section 3.1. We chose each component uniformly at random from six options: { AttentionBlock(128, 64), DenseBlock($R$, 128) for $R \in \{1, 2, 4, 8, 16\}$}. We sampled architectures with 13 layers each (for consistency with our primary model), and trained them on 5-way Omniglot. Averaged across 3 runs, these SNAILs achieved $98.62\% \pm 0.13\%$ and $99.71\% \pm 0.08\%$ for 1-shot and 5-shot, essentially matching the state-of-the-art performance of our primary architecture.

Finally, we explored the dependence of the classification strategy learned by SNAIL on the dataset it was trained on. If it truly learned an algorithm for few-shot classification, then a SNAIL trained on images from a particular domain should easily transfer to a new domain (such as between Omniglot and mini-Imagenet). To test this hypothesis:

- First, we took a SNAIL trained on 5-way Omniglot, fixed its weights, and re-learned an embedding for mini-Imagenet. Despite the SNAIL weights not being trained for mini-Imagenet, this method was able to achieve $50.62\%$ and $62.34\%$ on 1-shot and 5-shot.
- Then, we tried this in the reverse direction (freezing the SNAIL weights from mini-Imagenet, and re-learning an embedding for 5-way Omniglot), and this attained $98.66\%$ and $99.56\%$.
- Lastly, we combined an embedding trained on 5-way Omniglot with a SNAIL trained on 5-way mini-Imagenet (with a single linear layer in between, to handle the difference in feature vector dimensionality). We trained this model on 5-way Omniglot, where only the weights of the intermediate linear layer could be updated. It achieved $98.5\%$ and $99.5\%$.

All of these results are very competitive with the state-of-the-art, suggesting a strong degree of transferability of the algorithm and feature representation learned by SNAIL. An interesting idea for future work in zero-shot learning would be to learn embeddings for multiple datasets in an unsupervised manner, but with some mild distributional constraints imposed on the output feature representation. Then, one could train a SNAIL on one dataset, and have it transfer to new datasets without a single labeled example.

## C  REINFORCEMENT LEARNING

### C.1  MULTI-ARMED BANDIT AND TABULAR MDP ARCHITECTURES

For the $N$-timestep, $K$-arm bandit problem, the total trajectory length is $T = N$. For the MDP problem with $N$ episodes per MDP, it is $T = 10N$ (since each episode lasts for 10 timesteps).

For multi-arm bandits and tabular MDPs, we used the same architecture. First, we applied a fully-connected layer with 32 outputs that was shared between the policy and value function. Then the policy used: TCBlock($T$, 32), TCBlock($T$, 32), AttentionBlock(32, 32). The value function used: TCBlock($T$, 16), TCBlock($T$, 16), AttentionBlock(16, 16).

We found that removing the attention blocks made no difference in performance on the bandit problems, whereas SNAILs without attention could not learn to solve MDPs.

### C.2  CONTINUOUS CONTROL ARCHITECTURES

For each simulation locomotion task, the total trajectory length was $T = 400$ (2 episodes of 200 timesteps each). We used the same architecture (shared between policy and value function) for all tasks: two fully-connected layers of size 256 with tanh nonlinearities, AttentionBlock(32, 32), TCBlock($T$, 16), TCBlock($T$, 16), AttentionBlock(32, 32). Then the policy and value function applied separate fully-connected layers to produce the requisite output dimensionalities.

### C.3  VISUAL NAVIGATION ARCHITECTURES

Unlike the other RL tasks we considered, the observations in this domain include images. We preprocess the images using the same convolutional architecture as Duan et al. (2016): two layers with {kernel size $5 \times 5$, 16 filters, stride 2, ReLU nonlinearity}, whose output is then flattened and then passed to a fully-connected layer to produce a feature vector of size 256.

The total trajectory length was $T = 500$ (2 episodes of 250 timesteps each). For the policy, we used: TCBlock($T$, 32), AttentionBlock(16, 16), TCBlock($T$, 32), AttentionBlock(16, 16). For the value function we used: TCBlock($T$, 16), TCBlock($T$, 16).

### C.4  ADDITIONAL REINFORCEMENT LEARNING HYPERPARAMETERS

As discussed in Section 5.2, we trained all policies using trust-region policy optimization with generalized advantage estimation (TRPO with GAE, Schulman et al. (2015; 2016)). The hyperparameters are listed in Table 7. For multi-armed bandits, tabular MDPs, and visual navigation, we used the same hyperparamters as Duan et al. (2016) to make our results directly comparable; additional tuning could potentially improve SNAIL's performance.

Table 7:  The TRPO + GAE hyperparameters we used in our RL experiments.

| Hyperparameter | Multi-armed Bandits | Tabular MDPs | Continuous Control | Visual Navigation |
|---|---|---|---|---|
| Batch Size (timesteps) | 250K | 250K | 50K | 50K |
| Discount | 0.99 | 0.99 | 0.99 | 0.99 |
| GAE $\lambda$ | 0.3 | 0.3 | 0.97 | 0.99 |
| Mean KL | 0.01 | 0.01 | 0.01 | 0.01 |

## D    REINFORCEMENT LEARNING: ABLATIONS

Here we conduct a few ablations on RL tasks: we explore whether an agent relying only on TC layers or only on attention layers can solve the multi-armed bandit or MDP tasks from in Section 5.2.

First, we consider an SNAIL agent without attention layers (only TC layers, which amounts to a variant of the WaveNet architecture introduced by van den Oord et al. (2016a)).

When applied to the bandit domain, we found that this TC-only model performed just as well as a complete SNAIL. This is likely due to the simplicity of this task domain, as successful performance on bandit problems does not require maintaining a large memory of past experience. Indeed, many human designed algorithms (including the asymptotically optimal Gittins index) simply update running statistics at each timestep.

However, this model struggled in the MDP domain, where a more sophisticated algorithm is required. The results are in the table below (with those of a random agent, SNAIL, LSTM and MAML duplicated from Table 4 for reference). This agent's asymptotic suboptimality suggests that its ability to internalize past experience is being saturated.

Table 8:  Ablations of SNAIL in the MDP domain.

| $N$ | Method | | | | |
|---|---|---|---|---|---|
| | Random | LSTM | MAML | SNAIL | SNAIL, TC-only |
| 10 | 0.482 | 0.752 | 0.563 | **0.766 $\pm$ 0.001** | 0.616 $\pm$0.001 |
| 25 | 0.482 | 0.859 | 0.591 | **0.862 $\pm$ 0.001** | 0.684 $\pm$0.001 |
| 50 | 0.481 | 0.902 | – | **0.908 $\pm$ 0.003** | 0.699 $\pm$0.002 |
| 75 | 0.482 | 0.918 | – | **0.930 $\pm$ 0.002** | 0.726 $\pm$0.002 |
| 100 | 0.481 | 0.922 | – | 0.941 $\pm$ 0.003 | 0.728 $\pm$0.003 |

Next, we considered a SNAIL agent without TC layers (only attention). Due to the sequential nature of RL tasks, we employed the positional encoding proposed by Vaswani et al. (2017b). This model, which is equivalent to their Transformer architecture, could not solve the bandit or MDP tasks. In both domains, its performance was no better than random. To no avail, we experimented with multiple blocks of attention and multiple heads per block.

We hypothesize that this architecture's inadequacy stems from the fact that pure attentive lookups cannot easily process sequential information. Despite their infinite receptive field, they cannot directly compare two adjacent timesteps (such as a single state-action-state transition) in the same way as a single convolution can. The TC layers are essential because they allow the agent to locally analyse contiguous parts of a sequence to produce a better contextual representation over which to attend.

