# OpenReview forum: "A Simple Neural Attentive Meta-Learner"
_ICLR.cc/2018/Conference — Accept (Poster)_

### Official Review · AnonReviewer1 · 2017-11-22
**Unsure about novelty, would like to see analysis of model**

**Rating:** 6
**Confidence:** 3

**Review:**

The authors propose a model for sequence classification and sequential decision making. The model interweaves attention layers, akin to those used by Vaswani et al, with temporal convolution. The authors demonstrate superior performance on a variety of benchmark problems, including those for supervised classification and for sequential decision making.

Unfortunately, I am not an expert in meta-learning, so I cannot comment on the difficulty of the tasks (e.g. Omniglot) used to evaluate the model or the appropriateness of the baselines the authors compare against (e.g. continuous control).

The experiment section definitely demonstrate the effort put into this work. However, my primary concern is that the model seems somewhat lacking in novelty. Namely, it interweaves the Vaswani style attention with with temporal convolutions (along with TRPO. The authors claim that Vaswani model does not incoporate positional information, but from my understanding, it actually does so using positional encoding. I also do not see why the Vaswani model cannot be lightly adapted for sequential decision making. I think comparison to such a similar model would strengthen the novelty of this paper (e.g. convolution is a superior method of incorporating positional information).

My second concern is that the authors do not provide analysis and/or intuitions on why the proposed models outperform prior art in few-shot learning. I think this information would be very useful to the community in terms of what to take away from this paper. In retrospect, I wish the authors would have spent more time doing ablation studies than tackling more task domains.

Overall, I am inclined to accept this paper on the basis of its experimental results. However I am willing to adjust my review according to author response and the evaluation of the experiment section by other reviewers (who are hopefully more experienced in this domain).

Some minor feedback/questions for the authors:
- I would prefer mathematical equations as opposed to pseudocode formulation
- In the experiment section for Omniglot, when the authors say "1200 classes for training and 432 for testing", it sounds like the authors are performing zero-shot learning. How does this particular model generalize to classes not seen during training?

---

> ### Author Response · Authors · 2018-01-05
> **RE: Unsure about novelty, would like to see analysis of model**
>
> Please refer to our main response in an above comment that addresses the primary and shared questions amongst all reviewers. Here we respond to your specific comments.
>
> “The authors claim that Vaswani model does not incorporate positional information, but from my understanding, it actually does so using positional encoding. I also do not see why the Vaswani model cannot be lightly adapted for sequential decision making. I think comparison to such a similar model would strengthen the novelty of this paper (e.g. convolution is a superior method of incorporating positional information.”
>
> >>> Vaswani et. al. [3] add to their feature vector a representation of where the example is in the sequence (described in section 3.5 of [3]).  This method crucially does not perform any local comparisons where the embedding of a particular image is modified depending on the others is being compared against (which Matching Networks [2] found to be essential). For the final paper, we will conduct an ablation to show how much SNAILs performance degrades when TCs are replaced with this method. Preliminary experiments on the MDP problem using attention like in [3] (with the positional encoding mentioned above) performed marginally better than a random policy. We will include this ablation (the [3]-style model on RL tasks) in the final version of the paper.
>
> “In the experiment section for Omniglot, when the authors say "1200 classes for training and 432 for testing", it sounds like the authors are performing zero-shot learning. How does this particular model generalize to classes not seen during training?”
>
> >>> During test-time, for the 1-shot 5-way and 5-shot 5-way problems, the model is given 1 labeled example of each of the 5 selected test classes and 5 labeled examples of each of the 5 selected test classes respectively. Therefore it is not zero-shot. The set of 432 test classes are not seen during training.

---

### Official Review · AnonReviewer3 · 2017-11-28
**The paper involves intensive experiments for the proposed design paradigm; yet the emphasis of the contribution for its generalizability could be more clear on the generalization to reinforcement learning and lacks more theoretical/intuitional studies.**

**Rating:** 6
**Confidence:** 3

**Review:**

The paper proposes a general neural network structure that includes TC (temporal convolution) blocks and Attention blocks for meta-learning, specifically, for episodic task learning. Through intensive experiments on various settings including few-shot image classification on Omniglot and Mini-ImageNet, and four reinforcement learning applications, the authors show that the proposed structure can achieve highly comparable performance wrt the corresponding specially designed state-of-the-art methods. The experiment results seem solid and the proposed structure is with simple design and highly generalizable. The concern is that the contribution is quite incremental from the theoretical side though it involves large amount of experimental efforts, which could be impactful. Please see the major comment below.

One major comment:
- Despite that the work is more application oriented, the paper would have been stronger and more impactful if it includes more work on the theoretical side.
Specifically, for two folds:
(1) in general, some more work in investigating the task space would be nice. The paper assumes the tasks are “related” or “similar” and thus transferrable; also particularly in Section 2, the authors define that the tasks follow the same distribution. But what exactly should the distribution be like to be learnable and how to quantify such “related” or “similar” relationship across tasks?
(2) in particular, for each of the experiments that the authors conduct, it would be nice to investigate some more on when the proposed TC + Attention network would work better and thus should be used by the community; some questions to answer include: when should we prefer the proposed combination of TC + attention blocks over the other methods? The result from the paper seems to answer with “in all cases” but then that always brings the issue of “overfitting” or parameter tuning issue. I believe the paper would have been much stronger if either of the two above are further investigated.

More detailed comments:
- On Page 1, “the optimal strategy for an arbitrary range of tasks” lacks definition of “range”; also, in the setting in this paper, these tasks should share “similarity” or follow the same “distribution” and thus such “arbitrariness” is actually constrained.

- On Page 2, the notation and formulation for the meta-learning could be more mathematically rigid; the distribution over tasks is not defined. It is understandable that the authors try to make the paradigm very generalizable; but the ambiguity or the abstraction over the “task distribution” is too large to be meaningful. One suggestion would be to split into two sections, one for supervised learning and one for reinforcement learning; but both share the same design paradigm, which is generalizable.

- For results in Table 1 and Table 2, how are the confidence intervals computed? Is it over multiple runs or within the same run? It would be nice to make clear; in addition, I personally prefer either reporting raw standard deviations or conduct hypothesis testing with specified tests. The confidence intervals may not be clear without elaboration; such is also concerning in the caption for Table 3 about claiming “not statistically-significantly different” because no significance test is reported.

- At last, some more details in implementation would be nice (package availability, run time analysis); I suppose the package or the source code would be publicly available afterwards?

---

> ### Author Response · Authors · 2018-01-05
> **RE: The paper involves ...**
>
> Please refer to our main response in an above comment that addresses the primary and shared questions amongst all reviewers. Here we respond to your specific comments.
>
> “in general, some more work in investigating the task space would be nice. The paper assumes the tasks are “related” or “similar” and thus transferrable; also particularly in Section 2, the authors define that the tasks follow the same distribution. But what exactly should the distribution be like to be learnable and how to quantify such “related” or “similar” relationship across tasks?”
>
> >>> Measures of task similarity would certainly be useful in understanding how well we can expect a meta-learner to generalize. However, it remains an open problem and beyond the scope of our work -- our contribution is the proposed class of model architectures, which we experimentally validate on a number of benchmarks (where there is a high degree of task similarity, and thus potential for meta-learning to succeed) from the meta-learning literature.
>
> “On Page 2, the notation and formulation for the meta-learning could be more mathematically rigid; the distribution over tasks is not defined. It is understandable that the authors try to make the paradigm very generalizable; but the ambiguity or the abstraction over the “task distribution” is too large to be meaningful. One suggestion would be to split into two sections, one for supervised learning and one for reinforcement learning; but both share the same design paradigm, which is generalizable.”
>
> >>> Our formulation of the meta-learning problem is consistent with prior work, as one can see in MAML [1] and Matching Networks [2].
>
> “For results in Table 1 and Table 2, how are the confidence intervals computed? Is it over multiple runs or within the same run? It would be nice to make clear; in addition, I personally prefer either reporting raw standard deviations or conduct hypothesis testing with specified tests. The confidence intervals may not be clear without elaboration; such is also concerning in the caption for Table 3 about claiming “not statistically-significantly different” because no significance test is reported.”
>
> >>> The confidence intervals in Tables 1 & 2 are calculated over 10000 episodes of the evaluation procedure described in Section 5.1 (95% confidence). The statistical significance in Table 3 is determined by a one-sided t-test with p=0.05. We will make these clarifications in the final version of the paper.

---

### Official Review · AnonReviewer2 · 2017-11-28
**Novel architecture, good results**

**Rating:** 7
**Confidence:** 4

**Review:**

This work proposes an approach to meta-learning in which temporal convolutions and attention are used to synthesize labeled examples (for few-shot classification) or action-reward pairs (for reinforcement learning) in order to take the appropriate action. The resulting model is general-purpose and experiments demonstrate efficacy on few-shot image classification and a range of reinforcement learning tasks.

Strengths

- The proposed model is a generic meta-learning useful for both classification and reinforcement learning.
- A wide range of experiments are conducted to demonstrate performance of the proposed method.

Weaknesses

- Design choices made for the reinforcement learning setup (e.g. temporal convolutions) are not necessarily applicable to few-shot classification.
- Discussion of results relative to baselines is somewhat lacking.

The proposed approach is novel to my knowledge and overcomes specificity of previous approaches while remaining efficient.

The depth of the TC block is determined by the sequence length. In few-shot classification, the sequence length can be known a prior. How is the sequence length determined for reinforcement learning tasks? In addition, what is done at test-time if the sequence length differs from the sequence length at training time?

The causality assumption does not seem to apply to the few-shot classification case. Have the authors considered lifting this restriction for classification and if so does performance improve?

The Prototypical Networks results in Tables 1 and 2 do not appear to match the performance reported in Snell et al. (2017).

The paper is well-written overall. Some additional discussion of the results would be appreciated (for example, explaining why the proposed method achieves similar performance to the LSTM/OPSRL baselines).

I am not following the assertion in 5.2.3 that MAML adaption curves can be seen as an upper bound on the performance of gradient-based methods. I am wondering if the authors can clarify this point.

Overall, the proposed approach is novel and achieves good results on a range of tasks.

EDIT: I have read the author's comments and am satisfied with their response. I believe the paper is suitable for publication in ICLR.

---

> ### Author Response · Authors · 2018-01-05
> **RE: Novel architecture, good results**
>
> Please refer to our main response in an above comment that addresses the primary and shared questions amongst all reviewers. Here we respond to your specific comments.
>
> “The causality assumption does not seem to apply to the few-shot classification case. Have the authors considered lifting this restriction for classification and if so does performance improve?”
>
> >>> This was done in line with past work on meta-learning (such as Santoro et al. [5]) for a maximally direct comparison. In general, past work (and this work) consider such processing because it’s compatible with streaming over incoming data (relevant for future large scale applications) and it aligns well with future extensions on few-shot active learning (where the model sequentially creates its own support set by querying an oracle to label a chosen image from the dataset).
>
> “The depth of the TC block is determined by the sequence length. In few-shot classification, the sequence length can be known a priori. How is the sequence length determined for reinforcement learning tasks? In addition, what is done at test-time if the sequence length differs from the sequence length at training time?”
>
> >>> In some RL tasks, there is a maximum episode length, and we can choose the depth of the TC block accordingly (this true for all of the tasks considered in our paper). If the episode length is unbounded, or differs between training and test, we can simply choose a reasonable value (depending on the problem) and rely on the fact that the attention operation has an infinite receptive field. We can think of the TC block as producing “local” features within a long sequence, from which the attentive lookups can select pertinent information.
>
> “The Prototypical Networks results in Tables 1 and 2 do not appear to match the performance reported in Snell et al. (2017).”
>
> >>>  Snell et al. [4] found that using more classes at training time than at test time improved their model’s performance. Their best results used 20 classes at training time and 5 at test time. To make their results comparable to prior work, we reported the performance of Prototypical Networks when the same number of classes was used at training and test time (Appendix Tables 5 and 6 in their paper), as all of the methods we listed in Tables 1 & 2 might also benefit from this modification.
>
> “I am not following the assertion in 5.2.3 that MAML adaption curves can be seen as an upper bound on the performance of gradient-based methods. I am wondering if the authors can clarify this point.”
>
> >>> These set of experiments were conducted to demonstrate the advantage of having highly general meta-learners where the meta-algorithm is fully-learned, especially when it comes to task distributions with a lot of exploitable structure. These continuous control problems originally introduced by the MAML paper can be easily solved by the agent identifying which task it is in over the first couple timesteps and then proceeding to execute the optimal policy.  In the above comment, which we should reword, we meant to say that MAML’s performance demonstrates that gradient-based methods which take their update steps after several test rollouts are fundamentally disadvantaged compared to RNN-like methods for this problem.

---

### Public Comment · ~Pranav_Shyam1 · 2017-11-20
**Comparison with Attentive Recurrent Comparators**

Hi,

Very nice work! However, your results on Omniglot seem to be well within the error margins of results reported with Attentive Recurrent Comparators. I hope that you can consider citing the results in your future revisions.

---

> ### Author Response · Authors · 2018-01-05
> **RE: Comparison with Attentive Recurrent Comparators**
>
> Thank you! And apologizes for missing this prior work; we will make sure to add ARCs as a baseline for the next revision.

---

### Public Comment · (anonymous) · 2017-12-13
**Public Source Code**

The idea is novel the experiment results are good in a range of tasks.
However, some details of the model design and experiment implementations need to be clarified with the help of the source code.
Can you release the code for the public please?

---

> ### Author Response · Authors · 2018-01-05
> **RE: Public Source Code**
>
> We are actively working on cleaning up the code base and disentangling it from dependencies that can’t be made public. We hope to release the code on GitHub very soon.

---

### Author Response · Authors · 2018-01-05
**Main Response to Reviewers**


We wish to thank the reviewers for their thoughtful feedback!

Below, we respond to some general questions present in all reviews:

Even though few-shot learning is not inherently a sequential problem, there is merit to obtaining contextual information before the attention kernel is applied. For example, in Matching Networks [2], the authors pass the feature vectors through an LSTM before doing their attention operator. Without TCs or a LSTM, the embeddings on which we “attend” are myopic in the sense that each example is embedded independently of the other examples in the support set. The use of embeddings that are a function of the entire support set was found to be essential in [2].

In Appendix B Table 6, we conduct several ablation studies to evaluate how much each aspect of SNAIL contributed to its overall performance (in the few-shot classification domain). These suggest that both TC’s and attention are both essential components of the SNAIL architecture.

The primary strength of SNAIL is that it is maximally generic, making no assumptions about the structure of the learned meta-algorithm. In domains such as few-shot classification, there exists human intuition about algorithmic priors. Methods built around particular inductive biases (such as Matching Networks [2] or Prototypical Networks [4], which learn distance metrics) work reasonably well; SNAIL is versatile enough to match or exceed this performance.

However, the true utility of meta-learning is to learn algorithms for domains where there is little human intuition (which occurs in many RL tasks). Indeed, our experiments show that MAML [3], which uses gradient descent as the algorithmic prior, performs poorly. In contrast, SNAIL’s lack of algorithmic prior offers the flexibility to learn an algorithm from scratch. Furthermore, in domains such as bandits or MDPs, we show that SNAIL is competitive with human designed algorithms.

As discussed in the paper, one caveat is that SNAIL’s flexibility results in a large number of model parameters, which can lead to overfitting on extremely out-of-distribution tasks. For instance, the official Omniglot test set represents task distribution that are “close” to the training task distribution, and MNIST could define another task distribution that is much farther away from the training distribution. Following Matching Networks [2], we took a SNAIL model trained on [20-way, 5-shot Omniglot] and tested it in [10-way, 5-shot, MNIST], achieving 68% accuracy. In contrast, [2], which use distance metric as the algorithmic prior, obtains 72%.

Our analysis of thus: all meta-learning methods, given sufficient capacity, achieve roughly the same performance on the training task distribution (or when the distance between training and test distributions is zero). As this distance increases, any method’s performance will necessarily decrease; as indicated by our empirical results, SNAIL’s performance drops off slower than that of other methods, as SNAIL’s versatility allows it to learn more expressive and specialized strategies. However, as the distance from the training distribution becomes very large, in the regime where there is less potential for meta-learning, methods with algorithmic priors have slightly “heavier tails” than SNAIL. A benchmark that allows more precise extrapolation along this axis (distance from training distribution) would likely offer more insight, although the curation of such a benchmark is beyond the scope of this work.

Below, we respond to the individual comments of the reviewers.

[1] Finn et al. “Model Agnostic Meta-Learning”. https://arxiv.org/pdf/1703.03400.pdf
[2] Vinyals et al. ”Matching Networks for Few-Shot Learning”. https://arxiv.org/pdf/1606.04080.pdf
[3] Vaswani et al. “Attention is all you need”. https://arxiv.org/pdf/1706.03762.pdf
[4] Snell et al. “Prototypical Networks for Few-Shot Learning”. https://arxiv.org/pdf/1703.05175.pdf
[5] Santoro et al. “Meta-Learning with Memory-Augmented Neural Networks”. http://proceedings.mlr.press/v48/santoro16.pdf

---

### Decision · Program_Chairs · 2018-01-29
**ICLR 2018 Conference Acceptance Decision**

**Decision:**

Accept (Poster)

**Comment:**

An interesting new approach for doing meta-learning incorporating temporal convolution blocks and soft attention. Achieves impressive SOTA results on few shot learning tasks and a number of RL tasks. I appreciate the authors doing the ablation studies in the appendix as that raises my confidence in the novelty aspect of this work. I thus recommend acceptance, but do encourage the authors to perform the ablation experiments promised to Reviewer 1 (especially the one to "show how much SNAILs performance degrades when TCs are replaced with this method [of Vaswani et al.].")